# Spinal Muscular Atrophy Treatment in Patients Identified by Newborn Screening—A Systematic Review

**DOI:** 10.3390/genes14071377

**Published:** 2023-06-29

**Authors:** Karolina Aragon-Gawinska, Charlotte Mouraux, Tamara Dangouloff, Laurent Servais

**Affiliations:** 1Department of Neurology, Medical University of Warsaw, 02-097 Warsaw, Poland; karagon@wum.edu.pl; 2Neuromuscular Reference Center, Department of Pediatrics, University Hospital Liège, University of Liège, 4000 Liège, Belgium; charlotte.mouraux@chuliege.be (C.M.); tamara.dangouloff@uliege.be (T.D.); 3MDUK Oxford Neuromuscular Centre & NIHR Oxford Biomedical Research Centre, University of Oxford, Oxford OX3 0ER, UK

**Keywords:** spinal muscular atrophy, newborn screening, PCR, gene therapy

## Abstract

Background: In spinal muscular atrophy, clinical trial results indicated that disease-modifying treatments are highly effective when given prior to symptom onset, which has prompted newborn screening programs in growing number of countries. However, prognosis of those patients cannot be inferred from clinical trials conducted in presymptomatic individuals, as in some cases disease presents very early. Methods: we conducted a systematic review of articles published up to January 2023. Results: Among 35 patients with three *SMN2* copies treated before 42 days of age and followed-up for at least 18 months, all but one achieved autonomous ambulation. Of 41 patients with two *SMN2* copies, who were non-symptomatic at treatment initiation, all achieved a sitting position independently and 31 were able to walk. Of 16 patients with two *SMN2* copies followed-up for at least 18 months who presented with symptoms at treatment onset, 3 achieved the walking milestone and all but one were able to sit without support. Conclusions: evaluation of data from 18 publications indicates that the results of early treatment depend on the number of *SMN2* copies and the initial neurological status of the patient.

## 1. Introduction

Spinal muscular atrophy (SMA) is an autosomal recessive neuromuscular disease with reported incidence from 1:6000 to 1:30,000 [1,2,3]. The disease results from a homozygous deletion in the *SMN1* gene, which encodes the SMN protein that is critical for maintenance of motor neurons, in about 95% of cases [4]. The remaining cases come mostly from heterozygous deletion in one allele and a point mutation in the second allele. SMN protein functions in RNA splicing and in other cellular mechanisms such as axonal transport [5,6]. Humans have various numbers of copies of *SMN2*, which differs from *SMN1* by a mutation in the splice acceptor site of intron 7 [7]. As a consequence, only around 10% of SMN protein produced from *SMN2* is functional. In the case of total loss of function of the *SMN1* gene, the level of SMN protein produced from *SMN2* copies is inversely correlated with disease severity [8]. Thus, the *SMN2* copy number is the main, but not the only prognostic factor of SMA type.

Clinically, SMA always presents with progressive muscle weakness and atrophy but with a very wide spectrum of severity ranging from a severe prenatal form to an adult-onset form. Prior to the availability of genetic testing, SMA was classified according to the age of first manifestation and the highest motor milestone achieved [9]. Symptoms of SMA type 0 are observed in utero, and the patient presents with clear symptoms at birth. Symptoms of SMA type 1, the most common form [10], are observed within first 6 months of life, and patients never gain the ability to sit independently. SMA type 2 usually manifests after 6 months of life; children are able to sit independently but never walk independently. SMA type 3 patients acquire the ability to walk but ambulation can be lost as disease progresses. Loss of ambulation is earlier and more likely if the first symptoms appear in early childhood [11]. More than 50% of patients with symptom onset before 3 years of age, referred as type 3a, lose ambulation before adulthood. Approximately half of patients with symptom onset after the age of 3 years, classified as type 3b, are able to walk after the age of 40 [9,11]. Finally, symptoms of SMA type 4, also called “adult type”, appear in adulthood, and patients do not lose their ability to walk.

There is a clear correlation between SMA phenotype and the number of copies of *SMN2*. About 86% of patients with SMA type 1 have two copies of *SMN2*, and 87% of patients with SMA type 2 have three copies of *SMN2*. Of patients with SMA type 3, 64% have three copies of *SMN2* and 31% have four copies of *SMN2* [12]. Mutations in *SMN2* also modify the severity of SMA [13]. The most common is the c.859G > C variant, which promotes exon 7 inclusion, increasing the amount of correct SMN protein produced [13,14]. These modifiers account for milder phenotypes in patients with fewer copies of *SMN2*, but a minority of patients have these mutations. In addition, c.859G > C variant is rarely evaluated in current clinical practice. Classification based on symptom onset and SMA type is not appropriate for patients identified by newborn screening (NBS). In patients with no symptoms, *SMN2* copy number may provide an indication of when a patient will develop the first manifestations of the disease and what milestones will be achieved in the absence of treatment.

Although palliative care and respiratory support increased median survival, untreated patients with the most severe form of the disease usually do not survive past the age of 2 [15]. Patients with types 2 and 3 have close-to-normal life expectancy if properly managed [16]. In 2016, the first drug to treat SMA, an antisense oligonucleotide injected intrathecally (nusinersen, Spinraza^®^), was approved by FDA. This approval was followed by approvals of an intravenous gene therapy in 2019 (onasemnogene abeparvovec, Zolgensma^®^) and an oral therapy in 2020 (risdiplam, Evrysdi^®^). All three drugs have been [17,18,19] or are being (16) tested in presymptomatic patients younger than 42 days. These trials demonstrated that all patients with three *SMN2* copies achieved autonomous ambulation before the age of two years and that about half of the patients with two *SMN2* copies met normal motor milestones with the other half presenting with mild-to-moderate motor delay. These striking results demonstrated the necessity of diagnosing SMA as early as possible, which prompted NBS initiatives in several countries [2,20,21,22,23] and rapid extension around the world [24,25].

The first real-world evidence of the efficacy of early treatment became available soon after NBS for SMA was implemented. Studies from Germany, the US, and Belgium demonstrated the dramatic effects of early treatment as subjects treated after symptom onset have more severe delays in motor milestone acquisition. These studies also revealed that about 40% of patients with two copies of *SMN2* present with symptoms within the first month of life [1,23,26]. The prognosis of these patients cannot be inferred from clinical trials conducted in presymptomatic patients, as symptomatic patients were excluded from these studies. In order to define more realistic expectations in this group of patients, we conducted a systematic review of treatment results in patients identified via NBS.

## 2. Materials and Methods

### 2.1. Literature Search

The aim of this review was to pool data on patients with SMA detected through NBS and the outcomes of their treatment. We conducted two literature searches, one in November 2022 and one in January 2023, of PubMed and Embase using a Preferred Reporting Items for Systematic Reviews and Meta-analyses (PRISMA) checklist for systematic reviews (Figure 1). We first searched for keywords: “spinal muscular atrophy” and “newborn screening”. We included original, full-text articles that described treatment of patients identified by NBS and their treatment outcomes. We required that articles were published after January 2017 and were written in English, French, Spanish, or Polish. We extracted data on motor development of children treated with nusinersen, onasemnogene abeparvovec, or risdiplam.

### 2.2. Selection of Studies

After removing duplicates, abstracts were read by two researchers (KAG and TD) who selected articles for full-text review. These articles were then included or excluded according to predefined criteria. In case of any discordant decision, a consensus was reached through discussion. When more than one article reported on the same or a subset of a patient cohort, we chose the most recent one. We separately analysed the extracted data from clinical trials and real-world experience.

According to the PICO technique, we defined our population (P) as SMA patients identified by newborn screening, the intervention (I) as treatment with SMN-enhancing therapies (i.e., nusinersen, onasemnogene abeparvovec, or risdiplam), comparison (C) as subpopulations classified based on *SMN2* copy number and neurological status at baseline, and outcome (O) as motor development reported as motor milestones achieved.

## 3. Results

### 3.1. Study Selection Process

An initial search of literature in November 2022 identified 223 articles, and a search in January 2023 returned 5 additional works. After removing 78 duplicates, 162 articles were screened by title and abstract, and 46 were selected for full-text review. Of these, 18 articles met our criteria for inclusion in subsequent analysis. The main reason for exclusion was that no data were given on milestone development or follow-up. The flow chart of the study selection process is shown in Figure 1. Information on the 18 studies that met our criteria is listed in Table 1. Of these articles, 13 report real-world data; information on these studies is given in Table 2.

### 3.2. Results from Clinical Trials of Subjects Treated Presymptomatically

Several articles detail the results of clinical trials of nusinersen, onasemnogene abeparvovec, or risdiplam in 79 presymptomatic SMA infants [17,18,19,31]. Inclusion and exclusion criteria for the trials are summarized in Table 3. Data for at least one year of follow-up were available for 61 patients. All studies included children younger than 6 weeks of age, with gestational age specified in the criteria and a requirement that no symptoms of SMA were present immediately prior to dosing. The clinical trial data are summarized in Table 3 and Table 4.

All children were alive and none required invasive ventilation at the time articles were published. In the study that evaluated the safety and efficacy of nusinersen, four infants required respiratory intervention during the study, three remained on non-invasive ventilatory support, and three were supported by gastrostomy [17]. All of those patients harboured two copies of *SMN2*. In the studies that evaluated the safety and efficacy of onasemnogene abeparvovec in 29 subjects [18,19] and of risdiplam in 7 patients [31], none of the infants required ventilator or nutritional support at the time that the data were published.

When we considered all treated subjects, irrespective of therapy type, motor development varied significantly depending on the number of copies of the *SMN2* gene. Of the 28 children with three copies of *SMN2*, 24 (85%) developed normally and only mild motor delays were reported for the remaining 4 children. All 28 were able to walk independently. The 33 patients with two copies of *SMN2* had a more heterogenous evolution. All patients were able to sit independently, but only 23 (70%) did so before 9 months of age as defined by the World Health Organisation’s (WHO) developmental milestones’ window [39]. Twenty-three (70%) were able to walk independently, but only twelve did so before the age of 18 months. Importantly, the children continued to progress developmentally throughout the observation periods, which in some cases followed children to 5 years of age [27]. Treatment with all the drugs was generally well tolerated, and no major adverse events were reported.

### 3.3. Results from Subjects Identified by NBS

We identified 13 articles that provided follow-up results for 153 patients with different numbers of copies of *SMN2* who were identified by NBS programmes (Table 2). The results of these articles are summarized in Table 5 and Figure 2 and Figure 3. Some reimbursement and logistics problems were reported that delayed diagnosis or treatment [1,32,34]. False negative results and subsequent treatment delays were due to heterozygous mutations or point mutations of *SMN1* not detected by PCR [26], human error [32], or incorrect quantification of *SMN2* copy number [1,35]. Parental refusal was another reason for treatment delay or no treatment at all [1]. Patients with two copies of *SMN2* who were not treated died [1,32].

Treatment was administered to most children with two or three copies of *SMN2* at mean ages of 23 and 52 days, respectively. Patients with three copies of *SMN2* received treatment in 38 of 41 reported cases. For those treated before the age of 42 days (28/38), all but one presented with normal motor development. One patient had slight motor delay and started walking independently at the age of 19 months. Of the 77 patients with two copies of *SMN2*, 73 received treatment. Of treated patients, 37 had symptoms of SMA at the first treatment. After 17 months of mean observation (range of 2.5–32 months), four of these subjects had met age-appropriate milestones. Of the 36 subjects treated presymptomatically, 22 showed no delay in motor development at a mean age of 15 months (range of follow-up 1–28 months). In 13 subjects, clear motor delay was reported at a mean age of 15 months (range 10–24 months). For one presymptomatically treated patient, no follow-up data were available [32].

Three patients with one copy of *SMN2* were identified by NBS, and all had severe symptoms at birth. Two died without treatment. The other infant received risdiplam at 2.5 months of age while in intensive care. At 6 months of age, the last reported follow-up, he was ventilated invasively, unable to feed orally, and had minimal finger movement [3].

Many patients with four copies of *SMN2* were not treated in the first years of NBS programmes as the accepted recommendation for those subjects at that time was to monitor and treat at symptom onset [40]. A study published in 2021 reported that in Germany only 14% (2/14) of patients with four copies of *SMN2* had been treated [1]; in a 2022 report, however, 72% (13/18) of patients were treated [35]. Of five patients with four copies of *SMN2* who began treatment after symptom onset, all achieved and maintained independent walking, but some grade of proximal weakness remained at follow-up.

Among patients with two copies of *SMN2*, 8% (6/73) had nutritional support and 12% (9/73) used non-invasive ventilation at last follow-up. In those with three and four *SMN2* copies, none of the patients used respiratory or nutritional support.

## 4. Discussion

Early treatment for children diagnosed with SMA by newborn screening has dramatically changed the prognosis for these patients. As clearly established from natural history studies, patients, who harbour three, four, or five copies of *SMN2*, could walk in only 34% of cases [41]. Those treated after the symptoms rarely achieve independent walking, if this was not acquired before the onset of the disease [42]. When treated in the presymptomatic state of the disease, children with three or more *SMN2* copies achieve normal development in more than 90% of cases and the remaining patients have mild motor delay.

The situation prognosis is slightly different for patients with two copies of *SMN2.* When treated prior to symptom onset, these patients also have considerably better developmental trajectories than untreated patients or patients identified due to symptom onset [17,19], but their evolution is more heterogenous than that of patients with more copies of *SMN2*. During clinical trials, SMA symptoms were an exclusion criterion, and therefore about one-third of patients who were referred were ineligible for enrolment in a clinical trial. In real-world data of patients identified by NBS, 37 of 73 patients with two copies of *SMN2* had SMA symptoms at treatment initiation. This explains the discrepancy between clinical trial results and real-world data. It is worth noticing that there is probably no clear definition of “SMA symptoms” at a very early stage. Examination by physicians familiar with SMA will surely reveal more subtle abnormalities that are not likely to be detected by paediatricians with no specific neurologic training or SMA experience in the absence of a positive screening test. Currently, it is difficult to quantify such early SMA symptoms. Use of functional motor scales like CHOP-Intend (Children’s Hospital of Philadelphia Infant Test of Neuromuscular Disorders) could help to some extent in measuring and comparing the severity of symptoms presented by infants at the time of treatment.

Results from clinical trials conducted in presymptomatic babies cannot apply to the overall population of children identified by NBS, as the outcomes for patients who become symptomatic before treatment are much less favourable than for those treated presymptomatically. That difference in outcome confirms the urgency for rapid treatment of infants diagnosed with SMA, especially those with two copies of *SMN2* [36,43]. Non-invasive in utero diagnostics and subsequent prenatal treatment should be considered when sufficient experience has been gained in well-designed clinical trials.

This review provides evidence that patients with three *SMN2* copies and no symptoms present with an excellent functional prognosis. At the other end of the spectrum, patients with two copies and symptoms at treatment initiation are very likely to present with motor delay and ambulation cannot be guaranteed.

This review does not assess the question of treatment in patients with four copies of *SMN2*. Recommendations regarding treatment and management of these patients when identified by NBS have been a subject of discussion. In the first expert consensus statement in 2018 [40], only half the experts were supportive of early treatment, and therefore a wait and see strategy was recommended. Two years later, the working group published a revision in which 11 of the 12 experts on the panel recommended early treatment of children with four copies of *SMN2* [44]. Based on the limited available data, it is very likely that early treatment will ensure normal motor development. The rationale against early treatment in patients with four *SMN2* copies was mostly related to the high cost, the potential safety issues of therapies, and the societal burden for treatment of patients who are perceived as less severely affected, although most of them are at significant risk of losing ambulation. The finding of discrepancy in *SMN2* copy number quantification constitutes an additional rationale for early treatment of children with four copies of *SMN2* as some of these patients could actually be reported as having less than four copies with a different assay [1]. In addition, the strict follow-up necessary for implementation of a wait and see strategy is not always possible [45]. Further, the responsibility for a decision about whether or not to start treatment in the absence of symptoms is overwhelming to some parents. Much more data are needed to ground robust recommendations in these patients.

This review did not identify enough patients with one copy of *SMN2* to ground robust conclusions on their future outcome. As they are very likely to be clearly symptomatic at birth and require immediate ventilation and nutritional interventions, the use of disease-modifying therapies should be carefully discussed with parents.

Currently, therapies for SMA are very expensive [46]. Several pharmacoeconomic analyses of NBS have been published that provide support for screening for SMA in newborns [47,48,49,50], but these analyses often rely on the data generated by studies in presymptomatic patients and not in patients identified by NBS. Robust health economic analyses are essential to determine the value of treating patients identified by NBS and with more than one therapeutic intervention. This review could help to provide more precise assumptions in the modelling of the population of patients identified by NBS.

The incidence of SMA, calculated based on data from NBS programmes, is in agreement with that reported earlier based on symptomatic patients. NBS programmes in Germany, the USA, Belgium, Japan, and Australia have reported prevalence in the range of 1 in 6910 to 1 in 19,940 births (Table 2), with an average of 1:14,848. Studies based on symptomatic patients reported incidence of 1 in 12,000 births [10]. This means that nearly all patients with *SMN1* bi-allelic mutations become symptomatic, even though very rare asymptomatic cases were reported [51,52].

There are several limitations to our review. As SMA is a rare disease, the groups of patients in the publications identified are very small. Additionally, some patients were described in more than one publication (i.e., patients identified by an NBS programme who were subsequently enrolled in a clinical trial). Small groups of patients did not allow for comparisons between different therapies. Further, data collection and reporting were heterogenous, making the comparisons complicated. Only reports published in English, French, Spanish, or Polish were included. As any systematic review, our data are based on published articles, which can result in a bias towards more scientifically active centres. Finally, the number of *SMN2* copies and the presence of symptoms are strong prognostic factors but neither indicator is always reliable. Indeed, identification of symptoms mostly relies on the accuracy of clinical examination and could be defined differently by different physicians. In addition, the method of quantifying the number of *SMN2* copies is not standardised across sites [14,53]. In verification studies of 20 subjects, seven had fewer copies of *SMN2* than initially estimated [14,53]. Additional validation studies and standardisation are necessary in order to be able to fully and reliably pool data.

Despite these limitations, this review provides aggregated data for the prognosis of SMA patients identified by NBS, which could be helpful for physicians during their initial discussion with parents or during a patient’s follow-up to confirm that they have realistic expectations. Finally, it provides health economic models of NBS with more accurate data of incidence and functional outcomes.

## Figures and Tables

**Figure 1 genes-14-01377-f001:**
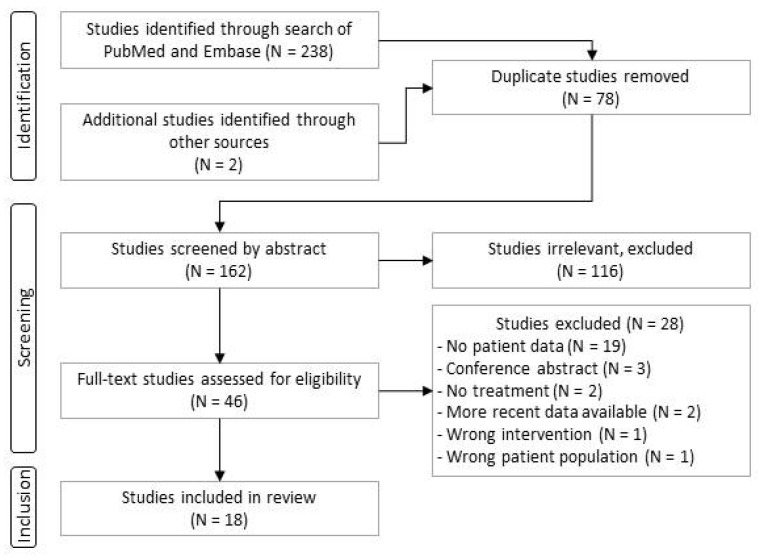
PRISMA flow chart for article selection.

**Figure 2 genes-14-01377-f002:**
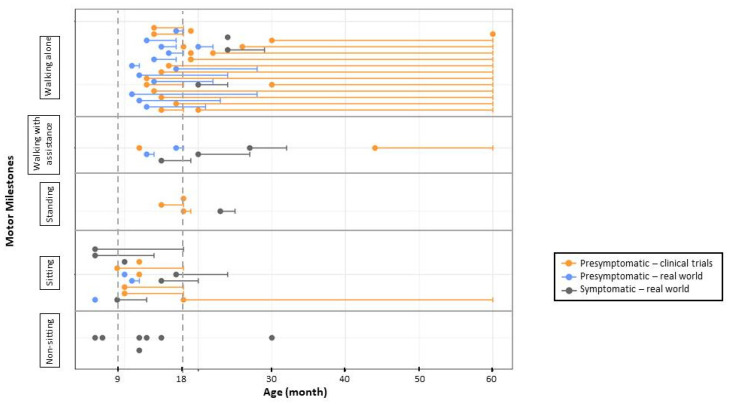
Motor development in patients with two copies of SMN2. Subjects presented were treated either in the context of clinical trials or in real-world use ≤42 days of life. Each circle or circle/line represents one patient. For each patient, data are shown in the row corresponding to the highest milestone reached. The position of the filled circle indicates the patient age at the time of the highest milestone achievement, the horizontal line illustrates follow-up time, and a short vertical line shows the age at the last follow-up visit.

**Figure 3 genes-14-01377-f003:**
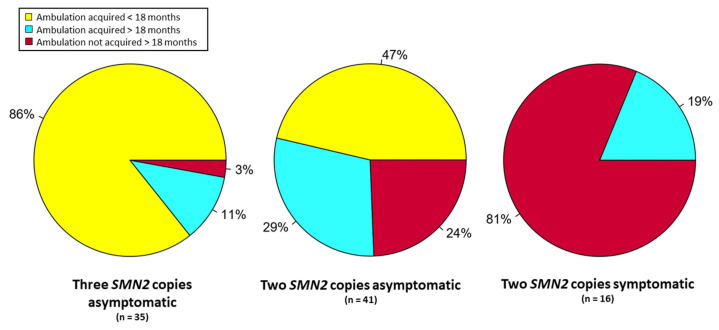
Ambulation in treated patients by *SMN2* copy number and SMA symptoms at treatment onset. Subjects were treated before 42 days of life and were followed for at least 18 months.

**Table 1 genes-14-01377-t001:** Articles reporting data on SMA patients treated through clinical trials or identified by NBS in chronological order by date of publication.

Article Author and Year [Ref.]	Location	Period	Study Description	*n* ^a^
DeVivo 2019, Kuntz 2022 [17,27]	USA, Australia, Canada, Germany, Italy, Qatar, Taiwan, Turkey	05.2015–02.2021	Preliminary results from NURTURE clinical trial of nusinersen	25 ^b^
Butterfield 2021 [28]	Utah, USA	2018	Two patients with two copies of *SMN2*	2
Kucera 2021 [29]	North Carolina, USA	10.2018–12.2020	Results from 2 years of NBS	1
Hale 2021 [30]	Massachusetts, USA	01.2018–01.2021	Results from 3 years of NBS	9
Boemer 2021 [26]	Southern Belgium	03.2018–02.2021	Results from 3 years of NBS	9 (10) ^c^
Vill 2021 [1]	Germany	01.2018–01.2020	Results from 2 years of NBS	42 (43) ^c^
Finkel 2022 [31]	USA, Australia, Belgium, Brazil, China, Poland, Russia, Taiwan	08.2019–07.2021	Preliminary results from RAINBOWFISH trial of risdiplam	7
Elkins 2022 [32]	Georgia, USA	02.2019–02.2021	Results from 2 years of NBS	15 (16) ^c^
Strauss 2022 [19]	USA, Australia, Belgium, Canada, Japan, UK	04.2018–12.2020	Final results from SPR1NT clinical trial of onasemnogene abeparvovec in patients with two copies of *SMN2*	14
Strauss 2022 [18]		04.2018–12.2020	Final results from SPR1NT clinical trial of onasemnogene abeparvovec in patients with three copies of *SMN2*	15
Lee 2022 [3]	New York, USA	10.2018–10.2021	Results from 3 years of NBS	32 (34) ^d^
Noguchi 2022 [33]	Hyogo, Japan	02.2021–08.2022	Results from 1 year of NBS	2
Matteson 2022 [34]	California, USA	06.2020–12.2021	Results from 1.5 years of NBS	16 (34) ^e^
Blaschek 2022 [35]	Germany	01.2018–01.2021	Outcomes in patients with four copies of *SMN2*	4 (20) ^d^
Schwartz 2022 [36]	Germany	01.2018–01.2021	Outcomes in patient with two copies of *SMN2*	6 (21) ^d^
Sawada 2022 [37]	Kumamoto, Japan	02.2021–01.2022	Results from 1 year of NBS	1
Kariyawasam 2023 [38]	Australia	08.2018–08.2020	Results from 2 years of NBS	14
			Total	214

^a^ N, number of patients included in present analysis, () number of total patients reported in the article. ^b^ Symptomatic patients excluded. ^c^ Patients not properly detected by NBS (i.e., false negatives) were excluded from analysis. ^d^ Patients were excluded from total cohort because of reported participation in clinical trial or inclusion within another cohort. ^e^ Limited follow-up data available only for 16 patients.

**Table 2 genes-14-01377-t002:** Articles reporting data from real-world NBS identification and treatment of SMA patients.

Study	Population	Number of *SMN2* Copies				Total Number of Subjects	Number of Subjects Treated	Number of Subjects Symptomatic at Treatment	Incidence
		1	2	3	≥4				
[26]	136,339	-	4	3	2	9	9	4	1:13,634
[32]	301,418	2	5	6	2	15	9	1	1:18,840
[30]	179,467	-	7	-	2	9	9	5	1:19,940
[29]	12,065	-	1	-	-	1	1	1	1:12,065
[28]	N/A	-	2	-	-	2	2	1	N/A
[3]	650,000	1	17 (18)	10 (11)	4	32 (34)	30	8	1:19,000
[34]	628,791	-	8	7	1	16	16	3	1:18,494
[33]	8336	-	2	-	-	2	2	1	1:25,000
[37]	13,587	-	-	1	-	1	1	0	1:13,587
[1]	297,163	-	17	9	16	42	26	6	1:6910
[36]	N/A	-	6 (21)	-	-	6 (21)	6	2	N/A
[35]	N/A	-	-	-	4 (20)	4 (20)	3	0	N/A
[38]	N/A		8	5	1	14	13	6	N/A
Total	2,227,166	3	77	41	32	153	127	38	1:14,848

In parentheses: total number of patients reported but who overlap with another included article.

**Table 3 genes-14-01377-t003:** Clinical trial inclusion and exclusion criteria.

Study	NURTURE	SPR1NT	Rainbowfish
Key inclusion criteria			
Age	<6 weeks	<6 weeks	<6 weeks
Gestational age (singleton)	37–42 weeks	35–42 weeks	37–42 weeks
Gestational age (twins)	34–42 weeks	35–42 weeks	34–42 weeks
*SMN2* copies	2 or 3	2 or 3	>1
CMAP	>1 mV	>2 mV	No limit
Weight	not specified	>2 kg and/or 3rd percentile	>3rd percentile
Key exclusion criteria			
Any signs or symptoms suggestive of SMA	At screening or immediately prior to the first dosing (Day 1)	At screening or immediately prior to dosing	At screening (or at baseline)
Respiratory	Hypoxemia < 96%	Hypoxemia < 96% (or <92% for altitude > 1000 m)	(SaO_2_ < 95%), requiring invasive ventilation, tracheostomy or awake non-invasive ventilation
Prior treatment	Any investigational drug or device, gene or cell therapy, or antisense oligonucleotide	Any investigational drug or device, gene or cell therapy, antisense oligonucleotide, drugs for treatment of myopathy, neuropathy, diabetes mellitus, immunosuppressive or immunomodulators, or plasmapheresis	Any investigational or commercial product, gene therapy, prior antisense oligonucleotide, *SMN2*-splicing modifier, oral β-2 adrenergic, or drugs with known retinal toxicity during pregnancy
Laboratory abnormalities	Clinically significant abnormalities in haematology or clinical chemistry	Clinically significant abnormalities of liver function tests (except for neonatal jaundice), blood count, AAV antibodies	Clinically significant abnormalities in laboratory test results
Other			ECG abnormalities

**Table 4 genes-14-01377-t004:** Summary of results from clinical trials.

	Study (Drug)	*N*	Mean Follow-Up (mo)	Follow-Up Range (mo)	Mean Age at Treatment (Days)	Age Range (Days)	Sitter < 9 Months	Sitter < 18 Months	Walker < 18 Months	Walker < 3 Years
Two copies of *SMN2*	NURTURE (nusinersen)	15	59	47–68	19	8–41	11 (73%)	15 (100%)	6 (40%)	13 (87%)
	SPR1NT (gene therapy)	14	18	18	20	8–34	11 (76%)	14 (100%)	5 (36%)	9 (64%) ^a^
	Rainbowfish (risdiplam)	4	12	12–15	26	16–40	1 (33%)	4 (100%)	1 (33%)	1 (33%) ^a^
	Total	33	36	12–68	22	8–41	23 (70%)	33 (100%)	12 (36%)	23 (70%) ^a^
Three copies of *SMN2*	NURTURE (nusinersen)	10	59	47–68	22	3–42	10 (100%)	10 (100%)	10 (100%)	10 (100%)
	SPR1NT (gene therapy)	15	24	24	32	9–43	11 (78%)	15 (100%)	11 (78%)	14 (93%) ^a^
	Rainbowfish (risdiplam)	3 ^b^	13	12–15	26	16–40	3 (100%)	3 (100%)	3 (100%)	3 (100%) ^a^
	Total	28	35	12–68	27	3–42	24 (86%)	28 (100%)	24 (86%)	27 (96%) ^a^

Abbreviation: *n*, number of patients, ^a^ Follow-up shorter than 3 years. ^b^ One patient was reported as having atypical 2–3 copies.

**Table 5 genes-14-01377-t005:** Data on individual treated patients identified by NBS.

	Ref	N	Mean Follow-Up (mo.)	Range Follow-Up (mo.)	Total Treated	Mean Age at Treatment (Days)	Age Range at Treatment Initiation (Days)	Symptoms at Treatment	Sitters (Age Range, mo.)	Walkers (Age Range, mo.)	Sitter at 9 Months	Sitter at 18 Months	Walker at 18 Months	Walker at 24 Months
**two** ** *SMN2* ** **copies**	[26] ^a^	4	21.5	14–32	4 (100%)	38	29–54	4 (100%)	4 (6–7)	0	4/4 (100%)	3/3 (100%)	0/4 (0%)	0/4 (0%)
	[30]	7	14	5–42	7 (100%)	21	11–38	5 (71%)	5 (7–15)	1 (12)	2/5 (40%)	3/3 (100%)	1/3 (33%)	1/1 (100%)
	[32] ^b^	5	2.75	2.5–3	3 (60%)	45	30–60	1 (33%)	n/a	n/a	n/a	n/a	n/a	n/a
	[28]	2	12	12	2 (100%)	22	20–24	1 (50%)	1 (9-)	1 (12)	1/2 (50%)	n/a	n/a	n/a
	[29]	1	3	n/a	1 (100%)	30	n/a	1 (100%)	n/a	n/a	n/a	n/a	n/a	n/a
	[3] ^c^	17	12	1–24	17 (100%)	35	12–89	8 (47%)	9 (6-n/a)	3 (n/a)	8/11 (72%)	3/4 (75%)	2/4 (50%)	2/2 (100%)
	[33]	2	4.5	3–6	2 (100%)	23	22–25	1 (50%)	n/a	n/a	n/a	n/a	n/a	n/a
	[36]	23	20	5.5–30	21 (91%)	22	14–39	8 (38%)	19 (6–17)	13 (11–24)	19/21 (90%)	14/15 (93%)	10/15 (66%)	4/8 (50%)
	[34]	8	12	n/a	8 (100%)	31	17–52	3 (38%)	n/a	n/a	n/a	n/a	n/a	n/a
	[38]	8	24	24	8 (100%)	27	19–36	5 (63%)	8 (n/a)	3 (n/a)	n/a	n/a	n/a	3/8 (38%)
	**Total**	**77**	**11**	**1–42**	**73** **(95%)**	**23**	**11–89**	**37** **(51%)**	**46** **(6–17)**	**21** **(11–24)**	**34/43** **(79%)**	**23/25** **(92%)**	**13/26** **(50%)**	**10/23** **(43%)**
**three** ** *SMN2* ** **copies**	[26] ^a^	3	23	12–33	3 (100%)	34	30–41	0 (0%)	3 (7-)	3 (11–15)	3/3 (100%)	3/3 (100%)	3/3 (100%)	2/2 (100%)
	[32] ^b^	6	7	4–14	6 (100%)	133	90–180	0 (0%)	2 (8–n/a)	n/a	n/a	n/a	n/a	n/a
	[3] ^c^	10	13	1.5–26	10 (100%)	37	11–94	0 (0%)	7 (n/a)	6 (11–n/a)	6/6 (100%)	2/2 (100%)	2/2 (100%)	2/2 (100%)
	[34]	7	12	n/a	7 (100%)	40	18–79	0 (0%)	n/a	n/a	n/a	n/a	n/a	n/a
	[37]	1	11	n/a	1 (100%)	42	n/a	0 (0%)	1 (n/a)	n/a	1/1 (100%)	n/a	n/a	n/a
	[1] ^d^	9	11	1.5–23	6 (66%)	24	20–29	0 (0%)	4 (7–n/a)	2 (12–19)	n/a	2/2 (100%)	1/2 (50%)	n/a
	[38]	5	24	24	5 (100%)	104	30–400	1 (20%)	5 (n/a)	5 (n/a)	n/a	5/5 (100%)	n/a	5/5 (100%)
	**Total**	**41**	**13**	**1.5–33**	**38 (93%)**	**52**	**11–400**	**1** **(2%)**	**22** **(7–n/a)**	**16** **(11–19)**	**10/10** **(100%)**	**12/12** **(100%)**	**6/7** **(86%)**	**9/9** **(100%)**
**four** ** *SMN2* ** **copies ^e^**	[26] ^a^	2	21	20–22	2 (100%)	44	39–49	0 (0%)	2 (5–6)	2 (12-)	2/2 (100%)	2/2 (100%)	2/2 (100%)	2/2 (100%)
	[30]	2	16	10–22	2 (100%)	90	8–171	1 (50%)	2 (7–n/a)	1 (12-)	2/2 (100%)	1/1 (100%)	1/1 (100%)	n/a
	[35]	18	25	12–44	13 (72%)	560	90–1440	5 (27%)	18 (n/a)	18 (n/a)	n/a	n/a	n/a	n/a
	[3] ^c^	2	8.5	7–10	1 (50%)	180	n/a	0 (0%)	2 (n/a)	n/a	1/1 (100%)	n/a	n/a	n/a
	**Total**	**24**	**23**	**7–44**	**18 (75%)**	**219**	**8–1440**	**6** **(25%)**	**24** **(5–n/a)**	**21** **(12–n/a)**	**5/5** **(100%)**	**3/3** **(100%)**	**3/3** **(100%)**	**2/2** **(100%)**

Abbreviations: mo., months; number of patients, NIV, non-invasive ventilation; n/a, not applicable or not available. ^a^ One heterozygote was excluded from this analysis. ^b^ One patient was not diagnosed in a timely fashion due to human error and was excluded from this analysis. ^c^ Two patients were enrolled in a clinical trial and were excluded from this analysis. ^d^ One patient with three copies of SMN2 was mistakenly diagnosed with four copies and was excluded from this analysis. ^e^ Patients reported in Refs. [30,32,36] were not considered as too little data were available.

## Data Availability

All data used in this article were published and/or reported.

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
