# Peer review of "Spinal Muscular Atrophy Treatment in Patients Identified by Newborn Screening—A Systematic Review"

_genes, 2023, doi:10.3390/genes14071377_

Round 1

Reviewer 1 Report

In this very clearly and comprehensively written systematic review the authors evaluate the outcome data of SMA patients identified by NBS and treated with disease-modifying treatments.- They conclude that the results of early treatment depend on the number of SMN2 copies and the initial neurological status of the patient. Patients with 3 SMN2 copies and no symptoms when treatment is started present with a very good functional prognosis. On the contrary, patients with 2 SMN2 copies and symptoms at treatment are very likely to present with a delay in motor development and may never reach ambulation.

This is an important overview of what has been published up to date about the motor outcome of patients with SMA identified by NBS and treated early with DMT. The manuscript includes data from the clinical trials and real-world experience with SMA NBS and early treatment, which makes the picture more complete and relevant to the field.

The manuscript is well-written and easy to read, but I would suggest to optimize the structure of the discussion (especially the first 4 paragraphs), the flow in that part is less fluent.

Small typo: line 248: This review provides (not provide).

The tables and figures are interesting, but the lay-out should be adapted for table 1: e.g. merge 'ref' and 'article author and year' columns; table 4: better in landscape; table 5: on one page.

Figure 2: why only patients with 2 SMN2 copies?

Figure 3: may be add 'asymptomatic' to 3 SMN2 copies?

Reviewer 2 Report

The manuscript titled: Spinal Muscular Atrophy treatment in patients identified by newborn screening – a systematic review, by Karolina Aragon-Gawinska et al attempts to compare the efficacy of treatments based on SMN2 copy number under conditions of clinical trials vs new born screening (NBS). 

The authors note a strong correlation of efficacy in patients with 3 copies of SMN2 in both the clinical trials and in subsequent real world data as a result of NBS—Perhaps as expected.  The authors focus on patients with 2 copies of SMN2 and real word efficacy as a result of NBS as compared to clinical trials.  The reason for this is that symptomatic patients, regardless of age of treatment, were excluded from clinical trials and therefore we really don’t understand what the prognosis would be of patients with 2 copies of SMN2 that are treated as a result of NBS, AND are also symptomatic. 

The results of the author’s analysis study are summarized in Figure 3.  They show that patients with 2 copies of SMN2 treated pre-symptomatically achieve a 76 % ambulation rate, whereas those treated after onset only achieve a 19% ambulation rate, even when treated within the same age range.  N=16.

There are several problems with this data. First, the method of data collection is based exclusively on the literature review. While there was seemingly careful attempts regarding inclusion criteria to not double count for patients and to discount ambiguous publications, still, the authors did not have direct access to patient data.  As a result, the authors could not effectively define “symptomatic” in terms of disease presentation, but only as a binary term.  In addition, there was no comparison of efficacy based on the individual therapeutic or dose.   The number of patients in groups are small (N=16)

While this reviewer understands the importance of the question the authors are trying to answer, it seems perhaps irresponsible to publish data sourced in this manner.  Especially, if the point of the manuscript is to be informative and predictive of treatments for a subset of patients.  It seems more relevant for this data to come from established Centers of Excellence and patient registries that can be more informative with respect to details, rather than speculative.  Have the authors tried to collaborate here?

All of the preclinical and clinical data point to early intervention for the best chances of successful treatment.  However, if we attempt to define the gray zone for patients with two copies of SMN2—it needs to be done on accessible patient data where “symptomatic” is defined for each patient and copy number is definitely assessed. 

Minor comments:

The tables are really difficult to digest.  Would suggest simplifying where possible.

Section 3.3 refers to Table 4.  Should this be Table 5?

Figure 2 is very confusing.  Needs better definition and explanation in supportive text

There is a lot of toggling between copy number and SMA Type.  Be consistent where possible.

Round 2

Reviewer 2 Report

I would strongly encourage follow up with Centers of Excellence to report on actual patients treated in the near future, rather than relying on mined literature.